# Yellow Pitahaya (*Selenicereus megalanthus* Haw.): The Less Known of the Pitahayas

**DOI:** 10.3390/foods14020202

**Published:** 2025-01-10

**Authors:** Daniel Valero, Alex Erazo-Lara, María Emma García-Pastor, Pedro Antonio Padilla-González, Vicente Agulló, Fátima Badiche El-Hiali, María Serrano

**Affiliations:** 1Deparement of Food Technology, EPSO-CIAGRO, University Miguel Hernández, Ctra. Beniel km. 3.2, 03312 Orihuela, Alicante, Spain; alex.erazol@espoch.edu.ec (A.E.-L.); ppadilla@umh.es (P.A.P.-G.); vagullo@umh.es (V.A.); fbadiche@umh.es (F.B.E.-H.); 2Escuela Politécnica Superior de Chimborazo (ESPOCH), Sede Morona Santiago, Macas 140101, Ecuador; 3Deparement of Applied Biology, EPSO-CIAGRO, University Miguel Hernández, Ctra. Beniel km. 3.2, 03312 Orihuela, Alicante, Spain; m.garciap@umh.es (M.E.G.-P.); m.serrano@umh.es (M.S.)

**Keywords:** pitahayas, quality, bioactive compounds, postharvest

## Abstract

Pitahaya or dragon fruit is an exotic fruit native to Mesoamerica and is cultivated in several regions of the world. In recent years, pitahaya has become increasingly in demand, firstly, for its good nutritional and organoleptic qualities and, secondly, for its richness in antioxidants and bioactive compounds. Spain has opted for new tropical crops, and among them, pitahaya is one of the most planted in recent years. Most of the investigations on pitahaya were conducted with red cultivars, while the research on yellow pitahaya (*Selenicereus megalanthus* Haw.) is very scarce. In this review, the current knowledge about types of pitahayas, the fruit growth and ripening, the quality attributes, the postharvest storage, the use of by-products, and the health attributes were covered.

## 1. Introduction

Pitahaya or dragon fruit is originally from Mexico and Central America, although it has also spread to other countries such as Brazil, Ecuador, Thailand, Vietnam, and lately Spain [1]. In recent years, the fruit has spread to more countries to which the cultivation of pitahaya can be adapted appropriately, since they have soil and climate characteristics similar to the places where it is originally from.

In recent years, pitaya has become increasingly in demand due to its good nutritional and organoleptic qualities, apart from its richness in numerous minerals, such as iron and phosphorus, in vitamins, mainly C and B, and in bioactive compounds [2,3,4]. The bioactive compounds have antioxidant properties and, in addition to preventing cellular aging, have an anti-inflammatory effect and stimulate collagen synthesis, among many other qualities [4]. On the other hand, pitahaya crop is easy and economical, which leaves a wide profit margin with its sale, since it is one of the most expensive fruits, reaching 25 EUR per kg on the market [2].

Pitahaya is a fruit that has been used in different ways since pre-Columbian times, for example, in food, dyes, and medicines. Fruits are living structures and, therefore, are affected by environmental conditions and the way they are handled. The lack of awareness about this aspect causes it to be exposed to inadequate temperatures and relative humidity, rough handling that causes impacts, cuts and compression, which accelerate the respiration and transpiration processes of the fruit, reducing its quality and shelf life [1].

Currently, the term pitahaya is used exclusively to designate the fruits of the *Selenicereus* and *Hylocereus* genera (Figure 1). Pitahaya and pitaya are exotic fruits that are often confused because of their similar names, but they currently belong to different plants. Pitahaya, also known as “dragon fruit”, comes from a type of cactus in the *Hylocereus* genus, while pitaya is from another cactus in the *Stenocereus* genus. The main difference between the two is their origin and appearance. For this reason, yellow pitahaya (firstly named as *Selenicereus megalanthus*) was re-classified as *Hylocereus megalanthus* [1].

The scientific information on the yellow pitahaya is much lower than that on the red pitahaya. The search in SCOPUS database of “pitaya” gave us 1375 scores, while it was 222 for “pitahaya”, and “yellow pitahaya” gave us only 48 hits. (www.scopus.com; accesed on 15 November 2024).

Spain has opted for new tropical crops, and among them, pitahaya has been one of the most planted. As can be seen in Figure 2, the south of the country is where most of it was planted: Andalusia, Murcia, the Valencian Community, and the Canary Islands. Before 2020 and based on data from MercaMadrid, dragon fruit was imported to Spain from countries such as Colombia, Ecuador, or Thailand. But, since 2020, data from national producers located in Alicante, Almería, Las Palmas, Murcia, and Santa Cruz de Tenerife began to be recorded, and in 2021 Huelva, Malaga, and Valencia were incorporated. Recently, associations and cooperatives of pitaya producers were created to give more visibility to the sector and create an official production and sales database. These associations are PitayaSpain, which groups together producers from Andalusia and is located in Malaga, and Pitapalma, which represents producers from the Canary Islands.

The aim of this review was to perform a review about yellow pitahaya, including growth cycle, fruit ripening, quality traits, nutritional characteristics, pre- and post-harvest factor affecting fruit quality traits, its health beneficial properties, and the potential uses of its by-products.

## 2. Economic Importance of Pitahayas

The tropical and exotic fruit sector in the EU has seen exponential growth in the last ten years, with an increase in imports of these fruits that can also be grown in garden greenhouses. Many of these fruits have become part of the daily consumption habits of the population and are increasingly occupying places on supermarket shelves. Pitahaya is one of these fruits. This fruit originates from the cactus of the *Hylocereus* genus, a climbing cactus plant that grows to great heights, with thick branches, and produces red or yellow fruits, which have about 20 different species [5].

In 2024, the size of the pitahaya market is USD 14.73 billion (EUR 13.34 billion) and is expected to reach EUR 16.55 billion by 2029. Three species of pitahaya are marketed internationally: *Hylocereus undatus*, which is the pitaya with red skin and white flesh, grown mainly in Vietnam, Thailand, Malaysia, Mexico, and Israel; *Hylocereus costaricensis*, with red skin and red flesh, grown mainly in Thailand, Malaysia, Nicaragua, and Israel. Both are known commercially as ‘Dragon fruit’. The third species is *Selenicereus megalanthus*, a pitaya with a yellow skin and white flesh, comprising 76.4% of commercial crops in Colombia [6].

## 3. Types of Pitahayas

Pitahaya comes from the family of cacti (Cactaceae), genus *Hylocereus*, and belongs to the species *H. undatus* and *H. megalanthus*, (syn. *Selenicereus megalanthus*), which are generally found in Latin America with lands ranging from a few meters to 1840 m above sea level (Table 1).

Historical data indicate that pitahaya is a word that comes from Haiti where it means “scaly fruit”; while other data from Mexico consider that the word is Quechua established by Spanish conquerors and refers to an edible fruit of the edible cactus, and in recent years, pitahaya is found in many parts of Mexico’ tropical forests [3]. On the other hand, Colombian authors claim that its origin is in their country and demonstrate it with their large exports of this fruit, but other researchers allude to its discovery in South America without specifying the nation [2,7].

The first species cultivated in Tenerife (Spain) were *H. undatus*, *H. triangularis*, and *H. hybridum* (red ones). Some of the other pitahaya species mainly cultivated around the world are described below.

*Hylocereus hybridum*: The fruit of this species is red on the outside and inside. It has a sweet and very pleasant flavor. Due to its characteristics, it is classified as productive and is sought after in export markets. It flowers from June to October, in summer or fall [1].

*Hylocereus costaricensis*: This species of pitaya is from Costa Rica, has a red peel and flesh color, is easy to grow, and grows fast. It grows in temperate climates with temperatures ranging from 10 °C to 35 °C. Fertile soil with good drainage should be used, in full sun or semi-shade exposures. It flowers from July to October, so extra light in early spring stimulates the production of flower buds [8].

In several Latin American countries, species from *Hylocereus* genus are planted on family lands, using traditional agronomic labor, selecting the fruit by themselves, and performing irrigation, where it involves basic technology, while other countries, such as Israel, have high sales of pitahaya due to the modern technology invested in agriculture that can obtain greater tons of crop per hectare. The yellow pitahaya (*Hylocereus megalanthus*, syn. *Selenicereus megalanthus*) is known internationally and mainly cultivated in Ecuador, Peru, Bolivia, and Venezuela [5,6,7]. In Ecuador, the two most important ecotypes are ‘Pichincha’ or ‘Nacional’, with an average weight of 150 g and ‘Palora’ with higher weight (350 g), and are cultivated in Palora Cantón in Morona Santiago [5,7].

Yellow pitahaya has thick and scaly skin (Figure 3) [6]. On the other hand, pitaya, found in more arid regions of America, has a thinner and thornier skin, and its flavor is sweeter and less watery than that of pitahaya. Although both pitaya and pitahaya are rich in nutrients and possess antioxidant properties, pitahaya is more common in international markets, standing out for its striking appearance and refreshing flavor. In India, the evaluation of both fruits revealed that the yellow was bigger (468 g) than the red one (367 g) [9].

## 4. Fruit Growth and Ripening

The introduction of high-value products, such as tropical or exotic fruit trees, into the greenhouse can increase the profitability of farms and contribute to diversifying agricultural production. In the case of pitahayas, both open air and greenhouse systems are used for growing pitahayas. The growth cycle includes flowering, single-pattern fruit growth and ripening. However, durations are different with 125 days for yellow pitahaya and 60–70 for red fruits [6,10].

The crop is mainly propagated vegetatively, by cladodes or stems, so clones are cultivated. The massive introduction of plant material in this format should be regulated and controlled to avoid the introduction of pathogens, such as the cactus virus X (CVX). In addition, it is necessary to evaluate its behavior in different latitudes and cultivation systems, since it is not possible to extrapolate the results obtained in other conditions [11].

It was reported that pruning has a great effect on crop yield. Thus, in a study leaving 6, 9, 12, or 15 cladodes per linear meter in pitahayas trained in a flat T system, the intensity of the pruning should be between 12 and 15 to obtain maximum flowering and fruit crop yield with high temperatures. At the end of the dry season, with high temperatures, buds sprout from the nodes, which are the size of a bean, and they take approximately 10 to 20 days to flower depending on the variety of pitahaya. Pitahaya flowers only open once at night and close in the morning, so pollination takes place in this short interval [6]. Thus, flowers open from 5:00 p.m. to 9:00 a.m. of the following morning, and the fruits ripen between 30 and 40 days after the flower is pollinated [3,6,12]. Flowering occurs from May to November and especially at the beginning and end of the period, when cross-pollination is poor, the fruit does not set [13].

Yellow pitahaya begins to bear fruit approximately 14 months after its final planting, from its transplantation. The fruit presents sigmoidal growth and requires approximately 13 to 14 weeks in the summer and 20 to 22 weeks in the winter for its development after pollination [10,14]. It is important to know the flowering cycles of the plant, since it allows the producer to plan their cutting or harvesting period. In the same plant, several fruit development phases can coincide at a given time: ripe fruit, fruit with 12–20 days of development, flowers about to open, flowers 2 days after flowering, and flower buds that have just begun. It is a high-yield plant, because while it bears fruit it continues to flower. Its production period is from May to September, and each fruit weighs 300–800 g depending on the cultivar [6,9,14]. The first harvest per red pitaya plant is more than 3 to 5 units, the yield in the first year of harvest is 4.5–6.5 tons per hectare, but from the third year, it enters its full production, when the yields are 11–12 tons per hectare [15].

Regarding its edaphoclimatic requirements, pitahaya is sensitive to low temperatures and does not tolerate frost. On the contrary, it tolerates high temperatures (some species up to 45 °C) and prefers medium–high relative humidity due to its tropical nature. Although it adapts well to a wide range of soils, it is sensitive to waterlogging, so clay soils should be avoided. In addition, it is tolerant to salinity and drought, but normally, depending on the rainfall received, it requires irrigation to maximize production (500–2500 m^3^ per hectare per year) [7,10,11].

Its annual cycle begins with the emission of new vegetative shoots at the end of winter, between February and April (vegetative phase). It then flowers in waves when the days are longer than the nights, as it is a long-day photoperiod species, and temperatures should exceed 18 °C, which generally occurs between May and November in our conditions. The fruit is harvested about 30–35 days after flowering, so the harvest is concentrated in the months of June to December, with more pronounced peaks in August and September. The fruits are characterized by the presence of very noticeable scales, in some cases, thorns with red or yellow skin and white, red, or fuchsia pulp, depending on the species and variety [6,12,13].

As for pests and diseases, some problems were detected with aphids, e.g., rot in the cladodes associated with bacteria, which appear in winter with low temperatures and high relative humidity, and the so-called “fish eye” that manifests itself with spots and chlorosis that appear on the cladodes caused by a fungus (*Botryosphaeria dothidea*) [16].

Fruit ripening was poorly studied in pitahaya, and determining the period to reach physiological maturity is essential to harvest pitahaya with the highest quality attributes [10,14]. Total soluble solids (TSSs), titratable acidity (TA), and firmness have been reported to be the most important quality attributes for yellow pitahaya and are related to its shelf life [2,9,10,14].

Kader (2002) described ripeness as the degree to which the fruit reaches its maximum development and presents sufficient quality for consumption. In addition, he indicated that a product has quality when it is in its optimal stage, which results from the combination of physical and chemical properties and characteristics that add value to the products [17]. Thus, there are different types of maturity which are as follows:Physiological Maturity. It is considered when the fruit reaches its maximum development, and the seeds are suitable for reproduction.Harvest Maturity. It is the physiological stage in which the fruit reaches consumption maturity and could be removed from the plant.Commercial Maturity. It is the way in which the fruit reaches the market and is accepted by the consumer. On the other hand, commercial quality is the physical presentation of the product and can be measured in different ways: by size, weight, color, shape, and shine and by the presence or internal and external defects, as well as those caused by pests.Maturity of Consumption. This is the stage in which the fruit has all the physical, chemical, and organoleptic characteristics. It must be taken into account that for non-climacteric products, ripeness for consumption is related to ripeness for harvest [17].

During on-tree fruit ripening, TSS increases in pitahaya fruit, reaching values ranging from 12 to 15 °Brix, depending on the cultivar, although in some cultivars, values as high as 20 °Brix have been reported in recently harvested fruits, while TA and firmness decreased [18]. The major sugars in red pitahaya are glucose and fructose, and malic acid is the main important organic acid [19], although no data are available in the literature about sugars and organic acid composition in yellow pitahaya.

## 5. Fruit Quality Parameters

As stated above, commercial maturity is reached when fruits meet the consumers’ required quality traits, having acquired the optimal level of sweetness and decreased acid and starch contents. Visually, this maturity stage can be assessed when the fruit changes from a green color to the typical color of the fruit (red or yellow). Once ripe, fruits can be harvested from the plant, or it can remain on the plant, increasing its weight for up to 50 days [20].

Ecuadorian exporters harvest Palora variety fruits for export when it has a greenish-yellowish cooler, with 15% ripeness [5], while the Colombian varieties are harvested with 50–75% yellowing [2]. The major quality parameters used as harvest indices include color changes in the peel, TSS, and TA, as well as the TSS/TA ratio defined as the ripening index (Table 2). TSS at harvest were more similar among pitahaya fruit species, ranging from 8.7 to 13.7 °Brix, than TA, which ranged from 0.15 to 1.7 g 100 g^−1^. Moreover, even for the same fruit species, 10-fold differences were reported, for instance, 0.15 and 1.47 g 100 g^−1^ for *H. megalanthus*, leading to ripening index values (TSS/TA) of 58 and 9. Thus, although, in these reports, the authors claimed that fruits were harvested at the commercial ripening stage, small differences among the harvest criteria between growers could lead to higher differences on sugar–acid ratio and, in turn, high differences in organoleptic properties. With respect to fruit size and weight, depending on the variety, the pitahaya fruit can measure between 8 and 12 cm long and 6 to 10 cm in diameter and can weigh up to 638 g (Table 1).

According to Le Bellec, et al., the red color of the skin and flesh of *Hylocereus* spp is due to betalains, a class of plant pigments, which are water soluble and synthetized from betalamic acid [25]. Among this pigment class, betacyanins were reported as the major pigments responsible for the reddish-violet color of pitahayas, and specifically, in *Hylocereus polyrhizus*, four betacyanins were identified by HPLC and LC-MS analysis, the major one being phyllocactin, followed by betanin and hylocerenin, while isobetanin was found at low concentrations [26]. On the other hand, for the yellow pitahaya, the peel color is due to the accumulated carotenoids [27,28] although there is no available literature regarding the specific carotenoid composition in pitahaya peel or pulp.

## 6. Preharvest Factors Affecting Pitahaya Quality

In this section, we examine the preharvest factors affecting fruit quality. With respect to temperature and radiation, despite being a cactus and having crassulacean acid metabolism (CAM) [29], pitahaya grows well in shaded places. It was reported that its optimal shade conditions are 35%, where it had a maximum net CO_2_ absorption rate, apart from an increase in the optimal stem length and width [30]. The water supply has to be between 600 and 1300 mm/year, although this depends on the physiological state of the plant, since, to stimulate flowering, this amount of water must be radically reduced for a period of 30 days [31].

Wild pitaya plants are quite versatile as they can grow in all types of soil, even in poor and stony soils in high mountains [32]. Even so, it is recommended that they be planted in sandy-loam soils with good drainage, rich in organic matter. The proportions of NPK fertilizers for the pitahaya crop depend on the phenological state of the plant. During initial growth, a high proportion of nitrogen is necessary to favor root and leaf growth, while during the reproductive period, fertilization with phosphorus (P) and magnesium (Mg) can also be added, which increases the synthesis of sugars in the fruits [31]. During this period when the fruit is setting, it is recommended to add a potassium fertilizer based on potassium oxide (KO_2_), since larger and better tasting fruits are observed, and the plant assimilates potassium very easily [33].

In modern agriculture, there is increasing interest in using natural compounds to enhance crop yield and increase market-desirable fruit quality characteristics. In this sense, the use of some plant growth regulators (PGRs) that mimic natural occurring plant hormones has shown to have interesting beneficial effects [34,35]. One of the aspects less studied in pitahaya is the effects of preharvest plant hormones and plant growth regulators in crop yield and performance as well as in improving quality traits. Pitahaya red or white flesh fruit species were classified as non-climacteric fruit, as ethylene has little or no effect on fruit ripening [31,36]. However, other plant hormones were shown to have some beneficial effects on fruit quality properties at harvest. For instance, the application of gibberellic acid at 0–500 mg L^−1^ to seedlings of white-fleshed pitahayas reduced the number of seeds at the time of harvest and during storage, and the best results were obtained with the highest dose [37]. In the last few years, the use of elicitor during fruit development has exponentially increased. Some of these elicitors are methyl jasmonate (MEJA), salicylic acid (SA), its derivative methyl salicylate (MESA), and melatonin (MEL), among others [38].

In red dragon fruit, (*Hylocereus polyrhizus* L.), the spraying of fruit with jasmonic acid (JA) and MEJA at 15 and 22 days after anthesis led to increased TSS [39]. However, in yellow pitahaya, no report has been published as of 2024. In a survey on the effect of several elicitors (MEJA, SA, and MESA) during growth and ripening, the results demonstrated that MESA advanced the ripening while MEJA delayed it, based on the highest and lowest values of TSS and TA, and increases in the highest crop yield were also observed. Thus, the preharvest application of MESA, MEJA, and SA at 1, 3, 5, and 10 mM at four different points of the cycle demonstrated that the highest yield [kg plant^−1^] was obtained with MEJA at 10 mM. Also, MESA showed the highest TSS, while MEJA showed the lowest. Overall, MESA advanced the ripening process, while MEJA delayed the pitahaya ripening process [10].

## 7. Postharvest Storage

The majority of research regarding pitahaya storage is dedicated to pitahaya red fruits, while very few reports exist on pitahaya during cold storage compared to yellow pitahayas. Pitahaya behaves as a non-climacteric fruit [3,31] that accumulates sugars during development, and it is very important to harvest the fruit at the optimal time of ripening based on the size and external color. On the contrary, if harvesting is delayed, the fruit turgidity decreases rapidly, and it becomes more sensitive to climatic conditions [31]. Pitahaya deteriorates relatively quickly under ambient conditions, with the temperature and storage period influencing their physiological processes and, in turn, affecting its shelf life. It is important to harvest in the early hours of the day to avoid an increase in the fruit’s temperature [40]. Care must be taken with the operations carried out at the time of harvest since these factors influence the quality of the fruit during post-harvest and at the time of marketing. This damage can be due to the used machinery, poor handling, and inappropriate transport temperature [40].

The postharvest storage life is affected by the rate of respiration and physiological weight loss. The recommended storage temperature for dragon fruit (*Hylocereus undatus* and *Hylocereus polyrhizus*) is 10 °C, and for yellow pitaya (*Selenicereus megalanthus*), it is 6 °C with 85–90% RH [14]. Even if pitahaya is stored at optimum temperatures, fruit quality deteriorates due to weight loss, softening by the pericarp’s cell wall degradation and reduction in starch soluble solids, monosaccharides, and malic acid content [41,42].

Pitahaya is very sensitive in developing CI symptoms, with the main effects being weight loss (WL), color changes, and loss of TSS and TA. CI induces some physiological defects affecting the market and storability of most tropical fruits. In recent years, it has been reported that fruits responded to chilling temperatures mainly in two ways. On the one hand, CI affects the membrane, changing its conformation and structure, with an increase in electrolyte leakage (EL) and lipid peroxidation, causing drastic fruit softening. On the other hand, low-temperature storage may induce an uncoupling in the respiratory chain and increase in reactive oxygen species (ROS) [41,43].

Although yellow pitahaya is characterized as a non-climacteric fruit, it responded to exogenous ethylene. Thus, the ripening of yellow pitahaya was induced by ethylene treatment, while treatment with 1-MCP (1-methylciclopropene), a blocking of ethylene action, delayed ripening and extended fruit shelf life [27].

Currently, the most important technologies employed for pitaya conservation are low-temperature storage combined with other technologies or treatments, such as modified atmosphere packaging, the application of chemical preservatives, coating, and the use of plant resistance inducers, such as melatonin or MeJA [36,44,45], although they have some limitations.

## 8. Health Attributes

Yellow pitahaya (*Selenicereus megalanthus* Haw) is reported as a good source of bioactive compounds, mainly polyphenols and vitamin C [46], and considered to exhibit health benefits since plant-dietary polyphenols show antioxidant activity [47]. Total phenolic content was reported to be very different depending on pitahaya fruit species and cultivars. Thus, Öziyci et al. [48] reported concentrations of TPC ranging from 38 to 621 to mg gallic acid/L in two white-fleshed and six red-fleshed *Hylocereus* sp. cultivars. In addition, it was reported that the phenolic content decreased from fruits set to commercial ripening in two dragon fruit species, *H. costaricensis* and *H. udantus* [49]. However, very few clinical studies were reported for yellow pitahayas, although beneficial health attributes were reported for red pitahayas mainly due to their bioactive components, including, apart from those previously mentioned, the betalain pigments. Betalains have gained significant interest in recent years for human diets due to the discovery that they are remarkably responsible for increased red-peel dragon fruit production because of their antimicrobial, antioxidant, antidiabetic, and anticancer effects [50,51]. For instance, extracts from *Hylocereus costaricensis* (red peel and pulp) and *Hylocereus undatus* (red peel and white pulp) were effective as cytotoxic activity toward colon and prostate cancer cells, with no toxic effect on normal cells [52]. In addition, extracts from red pitahaya (*Hylocereus polyrhizus*) were recently reported to exhibit antiviral effect against influenza A virus, which was attributed to betacyanins [53].

## 9. By-Products

Pitahaya is a fruit with high potential to produce new foods such as jams, jelly, yogurt, and wine, and approximately 33% of fruit production is processed. Thus, a large amount of by-products are generated during the agro-industrial processing of pitahaya since peels account as much as 51% of the whole fruit [54] (Figure 4).

Pitahaya peels are a by-product with valorization potential because it contains important macromolecules such as proteins, carbohydrates, mainly pectins with high purity soluble dietary and insoluble dietary fiber fatty acids; minerals; and bioactive compounds such as vitamins, betalains, phenolics, and terpenoids, among others, which could be used in the food or pharmaceutical industries [54,55,56]. Thus, the mucilage extracted from the peel of yellow pitahaya (*Selenicereus megalanthus*) fruit was characterized and determined the proximal analysis of proteins, lipids, crude fiber, ash, and carbohydrates, and the results showed that the mucilage from the peel was rich in fiber and carbohydrates [57].

The preharvest application of different elicitors to yellow pitahaya, particularly methyl salicylate (MeSa), methyl jasmonate (JaMe), salicylic acid (SA), and oxalic acid (OA), enhanced the content of polyphenols, carotenoids, macronutrients (K and Mg), and micronutrients (Fe and Zn) in the peel, especially MeSa [28], which is a good by-product.

In recent studies, the use of ultrasound-assisted extraction was reported as a good technique for extracting the bio-compounds from the peel of pitahaya [58], such as betalains, which have high antioxidant activity [39]. In addition, it has been reported that the peels contain higher amounts of betalains and radical scavenging capacity than flesh, although the concentration of betalains does not always correlate with the antioxidant capacity of the peel extract, showing the synergistic effects of other antioxidant compounds [59,60]. This highly rich betalain extract was recently used for incorporation into bioactive food packaging films with good mechanical and berry properties [61].

## 10. Conclusions

In this review, we demonstrated that yellow pitahaya has an excellent potential for new open markets, as demonstrated in its exponential increase in production in several countries, including Spain, due to its good adaptation to different soils and relatively stressed climatic conditions, such as those of the Mediterranean climate. From the point of view of fruit quality properties, yellow pitahayas have similar or even more TSS and total acidity TA contents, as well as antioxidant activity. In addition, the use of preharvest elicitors, such as MEJA, MESA, or oxalic acid, were reported as friendly, with no toxic tools used to increase crop yield and obtain fruits with higher quality properties at harvest and during postharvest storage. However, there is a need to search for more postharvest treatments or storage technologies to maintain fruit quality during long-term storage and investigate the use of by-products of yellow pitahaya peel as a source of polyphenols, nutrients, or minerals, since most of these studies were performed on red pitahayas.

## Figures and Tables

**Figure 1 foods-14-00202-f001:**
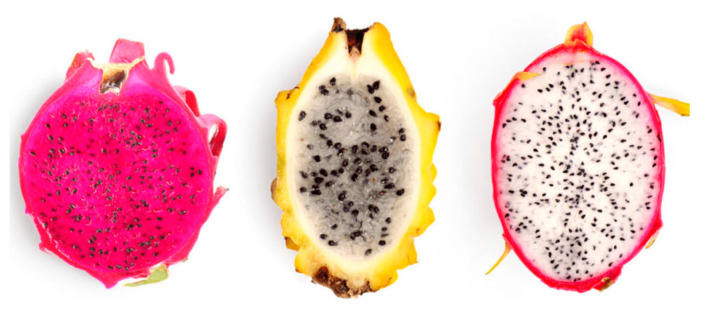
From left to right: Red pitahaya (*Hylocereus costaricensis*), yellow pitahaya (*Selenicereus megalanthus* Syn. *Hylocereus megalanthus*), and pitahaya rose (*Hylocereus undatus*). Source: https://www.freepik.es/ (accessed on 12 October 2024).

**Figure 2 foods-14-00202-f002:**
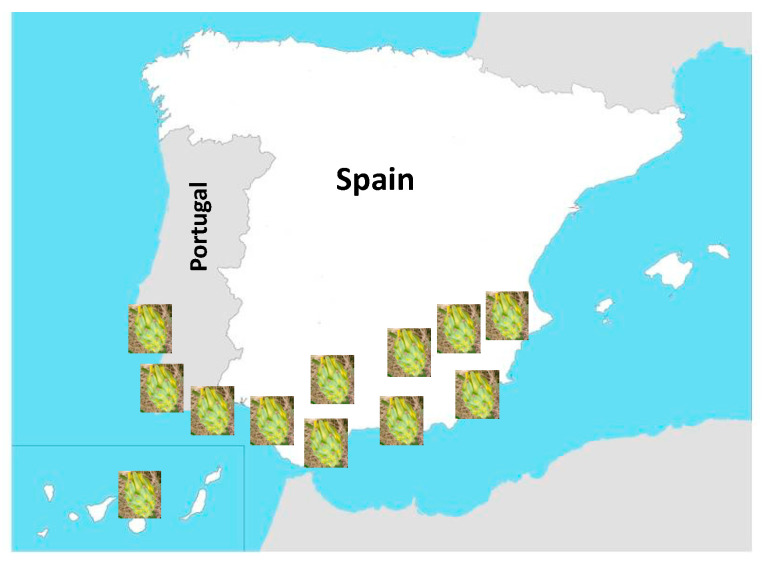
Distribution along the different provinces along Spain (source: the authors).

**Figure 3 foods-14-00202-f003:**
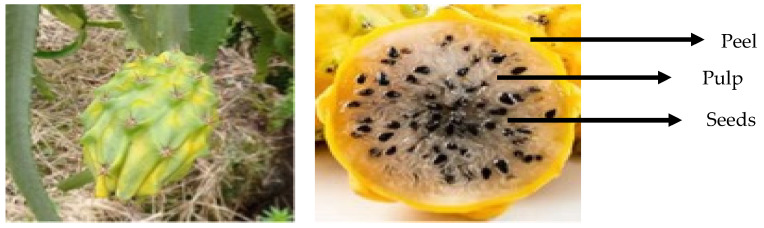
Immature fruit (**left**) and mature fruit (**right**) showing the yellow peel, white pulp, and black seeds. Photographs from the authors.

**Figure 4 foods-14-00202-f004:**
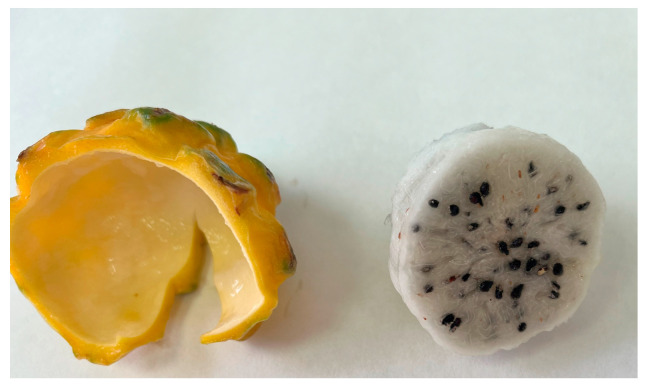
Peel and pulp of yellow pitahaya. Photograph from the authors.

**Table 1 foods-14-00202-t001:** Taxonomy of both types of pitahayas.

Yellow Pitahaya	Red Pitahaya
Kingdom: Plantae	Kingdom: Plantae
Division: Magnoliophita	Division: Magnoliophita
Class: Magnoliopsida	Class: Mognoliopsida
Order: Caryophillale	Order: Caryophillale
Family: Cactaceae	Family: Cactaceae
Tribe: Hylocereeae	Tribe: Hylocereeae
Genus: *Selenicereus*	Genus: *Hylocereus*
Specie: *Megalanthus*	Specie: *Undatus*
Category: Fruit	Category: Fruit
Scientific name: *Selenicereus megalanthus*	Scientific name: *Hylocereus undatus*
*(syn. Hylocereus magalanthus)*	

**Table 2 foods-14-00202-t002:** Quality parameters used as harvest indices for different pitahaya fruits for both red and yellow species ***.

	Weight (g)	Pulp (%)	Peel (%)	Color Skin	TSS (°Brix)	TA (%)	TSS/TA	Water (%)	Reference
** *H. megalanthus* **	430	51	49	Yellow	12.6	1.47	9	85.4	[10]
** *H. undatus* **	469	39	24	Red	12.6	0.41	33.3	88.3	[21]
** *H. polyrhizus* **	367	—	—	Red	13.7	0.24	15	89.3	[9]
** *H. undatus* **	—	—	—	Red	10.5	1.7	6.2	—	[22]
** *H. undatus* **	467	—	—	Red	13.2	0.24	76.2		[4]
** *H. megalanthus* **	347	65	32	Yellow	8.7	0.15	58.2	93.9	[23]
** *H. megalanthus* **	638	—	—	Yellow	12.6	0.48	23.7	—	[24,25]

* Total soluble solid (TSS), titratable acidity (TA).

## Data Availability

No new data were created or analyzed in this study. Data sharing is not applicable to this article.

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
