# Peer review of "Yellow Pitahaya (Selenicereus megalanthus Haw.): The Less Known of the Pitahayas"

_foods, 2025, doi:10.3390/foods14020202_

Round 1

Reviewer 1 Report

Comments and Suggestions for Authors

This review covers the current knowledge on the types, fruit growth and ripening, quality characteristics, postharvest storage, by-products, and health attributes of yellow pitahaya. It provides certain assistance in enhancing and deepening people's understanding of yellow pitahaya. However, there are still some issues with this article.

1.       Line 50: The first fruit in the image is commonly referred to as "red pitahaya," while the third fruit is labeled as "Pitahaya rose." Is the discrepancy between the figure caption and the image due to a specific variety name that does not align with our common understanding? Additionally, line 85 mentions that "Hylocereus undatus is the pitaya with red skin and white flesh," which indicates a mismatch between the figure and its description. Please examine this carefully.

2.       Missing Figure: There is no Figure 2 in the text, and it jumps directly from Figure 1 to Figure 3.

3.       Line 99, Table 1: Does "Phyllum" correctly represent the taxonomic rank of Phylum, which is positioned between Kingdom and Division? In the Class column, is "Magnoliopsida" a typographical error or a different variety compared to "Mognoliopsida"? The inconsistency between "red pitahaya (Hylocereus costaricensis)" in the table and the same description in the text causes confusion.

4.       Lines 129-133: These sentences present two points of confusion. Firstly, does Yellow pitahaya have three variants, or is it specifically referring to Yellow pitahaya with white flesh? Secondly, the comparison between "pitaya" and "pitahaya" in the sentence "On the other hand, pitaya, found in more arid regions of America, has a thinner, thornier skin, and its flavour is sweeter and less watery than that of pitahaya" is unclear.

5.       Figure 6: The caption for Figure 6 is incorrect.

6.       Figure 7: Has the image clarity met the required standards?

7.       Lines 323-332: The Health attributes of yellow pitahaya are not clearly presented.

8.       Line 337: Is it "restively" or "respectively"?

9.       Lines 343-349: The expression is unclear, and the sentence structure fails to clearly convey the main topic of yellow pitahaya.

Author Response

I UPLOAP A FILE

Reviewer 2 Report

Comments and Suggestions for Authors

Abstract

  • Line 19 ("que"): The word "que" is likely a placeholder or a translation remnant. If this is not intentional or part of a proper noun, it should be removed or clarified.

Introduction

  • Line 67: End the sentence as follows: “The following table shows the key characteristics of the samples analyzed.” Ensure clarity and completeness.
  • Line 72: The phrase "Also, we will use red pitahaya for comparative purposes" could be redundant if already mentioned. Consider rephrasing or removing the repetition. Specify if red pitahaya is a control or part of the study design to clarify its purpose.
  • Aim of the Study: Include a precise, concise aim in the introduction.

  • Figures and Tables
  • Figure 4: Replace this blurry figure with a clearer image at a higher resolution. Ensure it aligns with journal or document requirements.
  • Figure 5: Add a caption specifying the author or source of the photograph (e.g., *“Photo by [Author Name]” or “Image obtained from [source].”).
  • Figure 6 (Line 137): Verify this photograph's title and author/source. Correct errors in the title and add proper attribution.
  • Figures 7 and 8: Replace these blurry figures with higher-resolution images. Ensure they are properly sourced or attributed in the captions.

Citations and References

  • Lines 112, 118, 128, 161, 177, 184, 203, 252, 285, 328: Add references to substantiate these points. Ensure credible sources support all claims or data points.

Table 2

  • Clarify the abbreviations like TSS , TA, etc., in a footnote or legend to make the table self-explanatory.

Additional Points

  • Line 310 (HR): Define HR clearly in context. For instance,
  • Line 337 ("restively"): Likely a typo. Replace with the correct term, possibly “respectively.”
  • Lines 341–343: Provide more detailed data on the composition of the dragon fruit. Include numerical values such as nutrient content, pH levels, or other relevant parameters to add depth.

Conclusion

  • Rewrite the conclusion to include specific findings and future research directions.

Author Response

I UPLOAP A FILE

Round 2

Reviewer 1 Report

Comments and Suggestions for Authors

This manuscript provides a summary of some knowledge about the yellow pitaya. However, the revision work was not carried out carefully. No explanation or answer was given to comment and advice 6. The manuscript must be revised carefully before it can be published in this respected Journal.

Also, there were some details errors in the MS (eg. P399). Taking reference literature as an example, some have DOI numbers, while others do not. Also,the first letters of some references are still capitalized (22, 28, 38, 41, 56, 57). The authors should check again and modify them.

Author Response

I UPLOAD A FLE

Reviewer 2 Report

Comments and Suggestions for Authors

The paper has been changed correctly. 

Author Response

I UPLOAD A FILE
